# Vogt-Koyanagi-Harada Disease Exacerbation Associated with COVID-19 Vaccine

**DOI:** 10.3390/cells11061012

**Published:** 2022-03-16

**Authors:** Begoña De Domingo, Miguel López, Maria Lopez-Valladares, Esperanza Ortegon-Aguilar, Bernardo Sopeña-Perez-Argüelles, Francisco Gonzalez

**Affiliations:** 1Service of Ophthalmology, Instituto de Investigación Sanitaria (IDIS), Complejo Hospitalario Universitario de Santiago de Compostela, 15706 Santiago de Compostela, Spain; mariajesus.lopez.valladares1@rai.usc.es (M.L.-V.); francisco.gonzalez@usc.es (F.G.); 2NeurObesity Group, Department of Physiology, CIMUS, Instituto de Investigación Sanitaria, University of Santiago de Compostela, 15706 Santiago de Compostela, Spain; 3CIBER Fisiopatología de la Obesidad y Nutrición (CIBERobn), 28029 Madrid, Spain; 4Service of Neurology, Complejo Hospitalario Universitario de Santiago de Compostela, 15706 Santiago de Compostela, Spain; bdedbar@yahoo.es; 5Service of Internal Medicine, Complejo Hospitalario Universitario de Santiago de Compostela, 15706 Santiago de Compostela, Spain; bernardo.sopena@usc.es; 6Ophthalmology and Visual Science, CIMUS, University of Santiago de Compostela, 15705 Santiago de Compostela, Spain

**Keywords:** COVID-19 vaccine, VKH disease, vitiligo, uveitis, vaccine-associated autoimmunity

## Abstract

We describe a case of Vogt-Koyanagi-Harada (VKH) disease exacerbation after COVID-19 vaccination. A 46-year-old woman presented with a bilateral granulomatous uveitis 2 days after the first dose of COVID-19 mRNA vaccine (Comirnaty, Pfizer-BioNTech), and was diagnosed with a complete Vogt-Koyanagi-Harada (VKH) disease 4 days after receiving the second dose of the vaccine. Three weeks before the first dose, she had been consulted for blurred vision and mild headaches. The case resolved with high dose intravenous corticosteroids, followed by oral prednisone. The close temporal relationship between the COVID-19 vaccine doses and the worsening of VKH symptoms strongly suggests COVID-19 vaccination as the trigger of its exacerbation.

## 1. Introduction

The COVID-19 pandemic lead to the production and administration of anti-COVID-19 vaccines [1]. mRNA COVID-19 vaccines frequently lead to mild or moderate side effects [2,3,4]. However more serious side effects have been reported [5]. It has been suggested that the immunogenicity of mRNA COVID-19 vaccines could potentially exacerbate, induce or trigger autoimmune diseases [6,7]. Vogt-Koyanagi-Harada (VKH) disease is an autoimmune disease with systemic inflammatory manifestations including uveitis, meningitis, and auditory disturbances. Here we describe a case of a patient with VKH disease, which was exacerbated after the COVID-19 vaccine was administered.

## 2. Case Report

A 46-year-old woman consulted our Service of Ophthalmology in Santiago de Compostela, Spain, because of bilateral visual blurring and mild headache. Ten years earlier she underwent a LASIK procedure in both eyes but had no other relevant ocular history.

Three weeks later she received her first dose of mRNA COVID-19 (Comirnaty, Pfizer-BioNTech) vaccine and two days later she presented at the emergency service complaining of photophobia and worsening of her initial ocular symptoms. Best corrected visual acuity (VA) was 20/32 in both eyes, and intraocular pressure was 11 (OD) and 10 (OS) mmHg. Significant aqueous flare, keratic precipitates, iris nodules and cells in the anterior vitreous were present in both eyes. Poliosis was present. Pigment epithelium alterations were observed in the periphery of the eye fundus in both eyes (Figure 1).

Topical corticosteroids were prescribed and the patient was referred to the Uveitis Unit where a systemic examination revealed vitiligo and bilateral hypoacusis. Oral or genital ulcers were not present, neither signs nor symptoms of arthritis. Routine blood tests were unremarkable. ANA and ANCA were negative. ACE, ESR, lysozyme, and calcium serum levels were normal. Serology test for HBV, HIV, *Borrelia*, *Treponema pallidum*, *Rikettsia*, and *Mycobacterium tuberculosis* were negative. Her family history revealed a father with vitiligo, mother and grandmother with rheumatoid arthritis, and a brother with vitiligo and psoriasis. Topical corticoids were continued and a week later the VA recovered to 20/20 in both eyes, no aqueous flare was observed, but cells in the anterior vitreous were still present.

The patient received her second dose of COVID-19 vaccine 23 days after the first one, and four days later presented again at the emergency service complaining of visual loss, and severe headache that started two days earlier. The patient had impairment of her dysacusis. VA was 20/32 in both eyes. Aqueous flare, keratic precipitates, iridocapsular synechiae, and vitreous cells were observed in both eyes. Optical coherence tomography (OCT, Heidelberg Engineering, Spectralis-OCT) of the macula and the peripapillary area showed retinal and choroidal folds (Figure 2 and Figure 3). Choroidal thickening with no T-sign present was observed in the 20 MHz B-mode ocular echography. A further OCT (Topcon, Triton DRI-OCT) showed choroidal thickening and subfoveal neurosensory retinal detachment (Figure 4).

The patient was admitted to the hospital for study. Cerebrospinal fluid showed pleocytosis (57 cells/uL, 96% lymphocytes) and was negative for infection. Nasopharyngeal swab COVID-19 PCR test was negative. Onconeuronal antibodies were negative. Serum proteinogram was normal. Brain CT was normal but MRI showed some high intensity small spots located in the white matter of both frontal lobes and in the right parietal lobe. Differential diagnosis with sympathetic ophthalmia, primary B-cell lymphoma, posterior scleritis and uveal effusion syndrome was made and the condition was finally diagnosed as a complete VKH syndrome, following the revised Diagnostic Criteria for VKH disease [8]. Intravenous methylprednisolone (500 mg/day) for 3 days followed by oral prednisone was given. Two weeks later the condition improved. Ocular examination showed VA 20/20 in both eyes, no aqueous flare, some cells in the anterior vitreous, and some residual synechiae. Macular OCT, OCT-angiography, and fundus autofluorescence were normal, but the peripheral pigment epithelium alterations were still present.

## 3. Discussion

Because of the massive vaccination against COVID-19, it is possible to over-attribute many adverse effects to this vaccine. VKH disease results from an autoimmune process directed against melanocyte-associated antigens, which can be controlled when early and sustained corticoid/immunosuppressive treatment is followed [9]. Here, we describe the case of a patient with VKH disease whose exacerbation showed such a clear temporal relationship with the COVID-19 vaccine administration that it was likely this vaccine was the causative or triggering factor. mRNA vaccines, such as the one administered to our patient, generate the synthesis of viral proteins as early as day 1 post vaccination, which reaches a peak about 5 days later and then declines [10]. Our patient already had some prodromal symptoms before her first dose of vaccine, blurred vision and headaches, probably caused by a mild manifestation of her disease. Two days after she received the first dose of the vaccine her condition significantly worsened. This first episode of ocular inflammation was easily handled with topical steroids. Again, two days after she received the second dose, a more intense relapse of the disease occurred, this time meeting the criteria for a complete VKH disease. The lapse of time between the administration of the vaccine and the worsening of the condition was about two days on both occasions, and matches the time required for the expressed viral proteins to reach their peak. These proteins trigger the immune response, which in turn might activate the autoimmune activity responsible of the VKH disease. It must be noted however, that the temporal relationship we found may have been purely coincidental, given that in our region (Galicia, Spain) at the time this patient suffered the relapsing of her condition, about 70% of the population received at least the first dose of the COVID-19 vaccine.

Three cases of VKH disease have been described after SARS-CoV-2 vaccination [11,12,13]. Koong et al. [11] reported a case of VKH disease following the first dose of the Pfizer mRNA vaccine in a Chinese male without previous ocular history. They hypothesized that the vaccine could have triggered a common immune-mediated pathway that induced the disease. Papasavvas and Herbort [12] reported a case of VKH disease reactivation 6 weeks after the second dose of *Pfizer* anti-SARS-CoV-2 vaccination. The authors argued that the sudden recurrence of the disease after vaccination, when it had been well controlled during the previous 6 years, signaled the vaccine as a probable trigger, even after 6 weeks. Another case of complete VKH disease was described by Saraceno et al. [13] in a 62-year-old woman who had received the Oxford-AstraZeneca vaccine 2 days before the onset of symptoms. The authors also reported a case of VKH disease two weeks after having fever and anosmia and testing positive for SARS CoV-2 on a PCR test.

COVID-19 patients tend to develop multiple types of autoantibodies and there are other autoimmune diseases linked to the COVID-19 infection such as Gillian-Barre syndrome, Graves’ disease, autoimmune hemolytic anemia, thrombocytopenic purpura, systemic lupus erythematosus, myasthenia gravis and others [14]. The case we report here also suggests that VKH may be one of these diseases as well, and therefore should be considered in the vaccination protocols. Though COVID-19 vaccination does not induce *de novo* immune mediated adverse effects, it may trigger pre-existing latent autoimmune diseases [6]. Virus specific antibodies may enhance the entry of virus into monocytes, macrophages and granulocytic cells, a phenomenon known as antibody-dependent enhancement (ADE). A recent report remarked the potential risk of vaccine-induced ophthalmic ADEs [15], as could have occurred in our case.

In conclusion, to the best of our knowledge this is the fourth reported case of VKH disease reported after administration of a SARS-CoV-2 vaccine. Although it is difficult to determine causality, these cases raise the possibility of COVID vaccines being a trigger for VKH disease development, and this should be considered, especially in patients with vitiligo or previous history of the disease. Regardless, in all the reported cases, the patients responded to treatment and the benefit of vaccination probably outweighs the possible but very low risk of this side effect.

## Figures and Tables

**Figure 1 cells-11-01012-f001:**
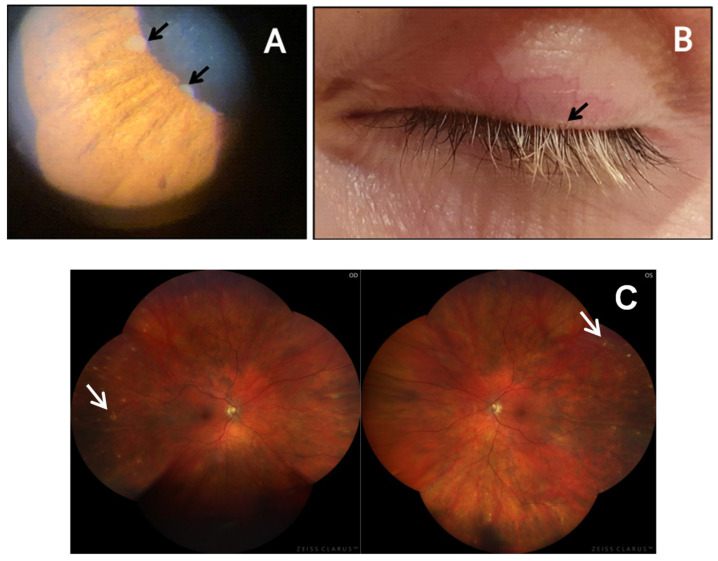
(**A**) Koeppe Nodules in the left iris (arrows). (**B**) Poliosis of the upper eyelashes of the left eye (arrow). (**C**) Pigment epithelium alterations (arrows) in the periphery of the eye fundus in both eyes.

**Figure 2 cells-11-01012-f002:**
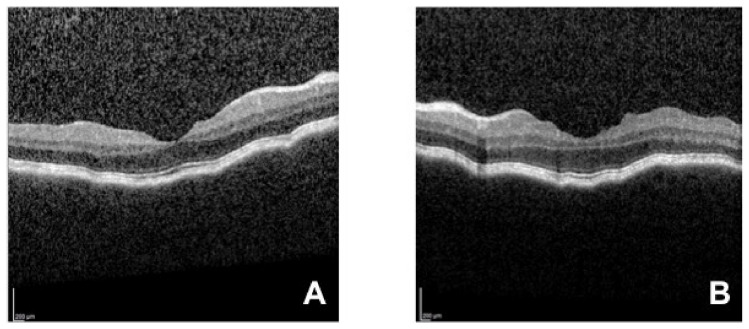
Right eye (**A**) and left eye (**B**) OCT of the macular area showing retinal and choroidal folds.

**Figure 3 cells-11-01012-f003:**
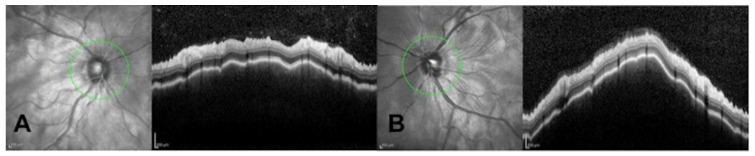
Right eye (**A**) and left eye (**B**) OCT of the peripapillary area showing retinal and choroidal folds.

**Figure 4 cells-11-01012-f004:**
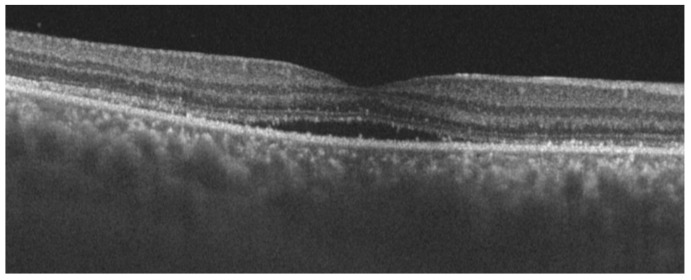
OCT image of the right eye, showing choroidal thickening and subfoveal neurosensory retinal detachment.

## Data Availability

The data presented in this study are available on request from the corresponding authors.

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
