# Peer review of "Vogt-Koyanagi-Harada Disease Exacerbation Associated with COVID-19 Vaccine"

_cells, 2022, doi:10.3390/cells11061012_

Round 1
Reviewer 1 Report
The report by Domingo et al., is an informative piece of work and will be of help to clinicians dealing with patient suffering from side-effects of COVID-19 vaccines. The report is precise and complete and ready for publication.
Author Response
Please, see attached file.

Reviewer 2 Report
I would like to thank the editor for the opportunity to review this manuscript
In this case report the authors are presenting a case of initial onset VKH disease in a patient previously vaccinated with a mRNA Anti-SARS-COV2 vaccine. Although there is no novelty as other cases have already reported there could be an interest in this case as the patient has presented signs after both doses of the vaccine.
These are my remarks
Major issues:
The authors mentioned that the diagnosis was a complete VKH disease based on the criteria of 2001. Although there are new revised criteria and other articles that propose new criteria for the diagnosis of initial onset VKH and chronic VKH, I will accept that the authors are based on those criteria. My major issue is that even with these criteria the authors are failed to demonstrate that the patient complete the 3rd criterium in order to call the disease a complete VKH disease. VKH is a stromal choroiditis though imaging of the choroid is very important in diagnosis. The authors are failed to demonstrate that the patient had a diffuse choroiditis or thickening of the choroid as they do not include in their case report the following:
1) They did not mention any Fluoresceine (FA) or more important any Indocyanine green angiography (ICGA) findings.
2) The OCT that presented is not an enhanced depth imaging (EDI) OCT, so we do not have any information of the choroid.
3) The OCT has demonstrated choroidal folds and retinal folds a finding that could be a posterior scleritis. No evidence of U/S in the article to exclude a scleritis
4) They mentioned that first episode of intraocular inflammation was managed with local corticosteroid drops, a treatment that is not sufficient for a VKH uveitis. Did the authors suppose that the first episode was another type of uveitis?
4) After treatment the authors mentioned that the condition was improved. Again, no mention of ICG angiography or FA.
Minor issues
1) I think that the authors should mention the family history of the patient after her personal history in order to have a complete history before entering the examination findings.
2) Authors describing iris nodules. They should mention if the nodule are Koeppe nodules or Busacca. The same at figure 1.
Author Response
Please, see attached file.

Reviewer 3 Report
Authors described a case of Vogt-Koyanagi-Harada (VKH) disease exacerbation after mRNA Covid-19 vaccine. The patient presented with bilateral granulomatous uveitis with vitiligo. Vitiligo usually occurs in the convalescent phase of the Vogt-Koyanagi-Harada (VKH) disease. The presence of vitiligo is quite unusual at the presentation. Vogt-Koyanagi-Harada (VKH) disease requires oral steroid or intravenous steroid. It is quite surprising that the patient had responded with topical corticosteroid alone in the first visit.
After a second dose of Covid-19 vaccine 23 days after the patient had came with complain of visual loss and severe headache. Exacerbation of VKH was diagnosed. That time the patient received the intravenous methylprednisolone for three days. Authors did not do any ultrasound and fundus fluorescein angiogram which is required for diagnosis of Vogt-Koyanagi-Harada (VKH). Diffuse posterior choroidal thickening is seen in the ultrasound in Vogt-Koyanagi-Harada (VKH) disease. Fundus fluorescein angiogram initially shows pinpoint hyper-fluorescence with pooling of the dye in the subretinal space in late phase. This has not been done. What kind of optical coherence tomography was done has not been mentioned. In case of Vogt-Koyanagi-Harada (VKH) disease, enhanced depth imaging optical coherence tomography usually show choroidal thickening which has not been done. There are some cases of Vogt-Koyanagi-Harada (VKH) disease described following SARS-CoV-2 vaccination.
The ancillary test imaging in this case is not appropriate particularly fundus fluorescein angiogram and ultrasound should have not been done.
Author Response
Please, see attached file.

Round 2
Reviewer 2 Report
Although the authors have responded in some comments, unfortunately they had not provide any dye exam to support the diagnosis of this uveitis. VKH is a stromal choroiditis and visualisation of the choroid with ICGA is essential to the diagnosis. Under this circumstances, I regretfully cannot recommend the manuscript for publication in its current form.
Reviewer 3 Report
I am satisfied with the revision made